# Unveiling the Anti-Cancer Potential of Onoceranoid Triterpenes from *Lansium domesticum* Corr. cv. *kokosan*: An In Silico Study against Estrogen Receptor Alpha

**DOI:** 10.3390/ijms241915033

**Published:** 2023-10-09

**Authors:** Ari Hardianto, Sarah Syifa Mardetia, Wanda Destiarani, Yudha Prawira Budiman, Dikdik Kurnia, Tri Mayanti

**Affiliations:** 1Department of Chemistry, Faculty of Mathematics and Natural Sciences, Universitas Padjadjaran, Jatinangor 45363, West Java, Indonesia; 2Research Center for Molecular Biotechnology and Bioinformatics, Universitas Padjadjaran, Bandung 45363, West Java, Indonesia

**Keywords:** estrogen receptor alpha, in silico ADMET, *Lansium domesticum* Corr. cv. *kokosan*, molecular docking, molecular dynamics simulation, onoceranoid triterpenes

## Abstract

Breast cancer is a significant global concern, with tamoxifen, the standard treatment, raising long-term safety issues due to side effects. In this study, we evaluated the potential of five onoceranoid triterpenes from *Lansium domesticum* Corr. cv. *kokosan* against estrogen receptor alpha (ERα) using in silico techniques. Utilizing molecular docking, Lipinski’s rule of five, in silico ADMET, and molecular dynamics simulations, we assessed the potency of five onoceranoid triterpenes against ER*α*. Molecular docking indicated competitive binding energies for these triterpenes relative to the active form of tamoxifen (4OHT) and estradiol, an ER*α* native ligand. Three triterpenes met drug-likeness criteria with favorable ADMET profiles. Notably, **2** demonstrated superior binding affinity in molecular dynamics simulations, outperforming estradiol, closely followed by **3** and **4**. Hierarchical clustering on principal components (HCPC) and the spatial distribution of contact surface area (CSA) analyses suggest that these triterpenes, especially **2**, may act as antagonist ligands akin to 4OHT. These findings highlight the potential of onoceranoid triterpenes in treating ERα-related breast cancer.

## 1. Introduction

Breast cancer is the most prevalent cancer in women around the world [1,2]. It is a major cause of death among women, significantly impacting global health. In 2020 alone, 2.3 million new cases of breast cancer were diagnosed, and 685,000 deaths were attributed to the disease [1]. Breast cancer incidence and mortality rates vary across different regions and countries, with higher rates observed in populations with high socioeconomic status [3].

Breast cancer has various subtypes characterized by different criteria, including pathological type, lymph node involvement, tumor size, and molecular subtypes [4]. The most recent classification is based on receptor status, including the expression of progesterone receptor (PR), estrogen receptor (ER), and human epidermal growth factor receptor 2 (HER2). Most breast cancer cases are ER-positive, accounting for approximately 70% of cases. ERs have three subgroups: ER*α*, ER*β*, and membrane receptors such as G protein-coupled estrogen receptor 1 (GPER1) [5].

ERs, including ER*α* and ER*β*, are activated by estrogens such as estradiol, forming homodimers or heterodimers with other ER-ligand complexes [6]. These dimers then activate the transcription of specific genes containing estrogen response elements (EREs) [7]. ER*α* is the predominant subtype in breast cancer, accounting for approximately two-thirds of cases, and can be modulated by agents like tamoxifen [8].

Tamoxifen, a selective ER modulator, is commonly used to treat ER-positive breast cancer by blocking the ER pathway [9]. However, it has several dangerous side effects [10,11,12,13,14,15,16]. These side effects include thromboembolism, nonalcoholic fatty liver disease (NAFLD), increased risk of endometrial hyperplasia or endometrial cancer, and venous thromboembolic disease [11]. Tamoxifen has been reported to contribute to the development of fatty liver, which can reduce drug compliance and increase the incidence of other metabolic diseases [13]. Other side effects attributed to tamoxifen include hot flashes, vaginal bleeding, gynecologic symptoms, depression, forgetfulness, sleep alterations, weight gain, and diminished sexual functioning [11].

Natural products have gained attention as potential alternative agents for combating breast cancer [17,18,19,20,21,22,23,24,25]. According to our previous research [26,27], secondary metabolites from *Lansium domesticum* Corr. cv. *kokosan* exhibit cytotoxic activities against MCF-7 breast cancer cells, which overexpress ER*α* [28,29,30]. Some onoceranoid triterpenes have been discovered from *L. domesticum* Corr. cv. *kokosan*. They include lansiolic acid (**1**) [31], 8,14-secogammacera-7,14-dien-3,21-dione (**2**) [32], 8,14-secogammacera-7,14(27)-dien-3,21-dione (**3**) [32], kokosanolide B (**4**) [27], and 3-hydroxy-8,14-secogamacera-7,14-dien-21-one (**5**) [33] (Figure 1). Onoceranoid triterpenes from other plants also possess cytotoxic activities against breast cancer, for example, reinerein A and lansic acid from the stem bark of *Reinwardtiodendron cinereum* [34] and lamestikumin A from the fruit peel of *L. domesticum* Corr. [35]. Therefore, onoceranoid triterpenes from *L. domesticum* Corr. cv. *kokosan* may also inhibit breast cancer.

In this study, we evaluated the anti-breast cancer activities of five onoceranoid triterpenes from *L. domesticum* Corr. cv. *kokosan* against ER*α* through in silico approaches. Molecular docking was performed as the initial stage to evaluate the anti-breast cancer activities of onoceranoid triterpenes. Subsequently, we employed Lipinski’s rule of five (Ro5) to predict the drug-likeness properties of the onoceranoid triterpenes. Additionally, we assessed the pharmacokinetics and toxicity of the screened onoceranoid triterpenes using in silico ADMET (absorption, distribution, metabolism, excretion, and toxicity). Ultimately, we subjected the onoceranoid triterpenes to 500 ns molecular dynamics simulations to better understand the inhibition of onoceranoid triterpenes against ER*α*. 

## 2. Results

### 2.1. Molecular Docking

The first stage in this study was molecular docking, as the initial evaluation to uncover anti-breast cancer activities of onoceranoid triterpenes from *L. domesticum* Corr. cv. *kokosan* against ER*α*. Before subjecting all onoceranoid triterpenes to molecular docking, we performed a redocking procedure using the complex of ER*α* and 4-hydroxy tamoxifen (4OHT) with a PDB code, 3ERT. Such a procedure warrants the finding of global minimum using the stochastic search method during molecular docking, which eventually minimizes errors [36]. Redocking such an active form of tamoxifen to ERα produces a hundred poses. Of a hundred poses of 4OHT, ninety-nine can reproduce the crystal structure conformation with RMSD values below 1.50 Å (Appendix A). The pose with the lowest RMSD value (0.66 Å) has a binding energy score of −11.03 kcal mol^−1^ (Appendix A), whereas the one with the most substantial binding energy score (−11.50 kcal mol^−1^) (Table 1) possesses an RMSD value of 1.14 Å (Appendix A). These results indicate that molecular docking validation was successful since the RMSD threshold in a redocking procedure is 2.00 Å [37]. 

Tamoxifen is an antagonist ligand for ERα, which disrupts the activity of the enzyme in cancer cells [38], while the native ligand for ERα is estrogens, such as estradiol [30]. The binding of estrogens to the ER*α* facilitates cancer cell proliferation through complex molecular signaling pathways [39]. Therefore, the current study included estradiol as the agonist ligand reference. The resulting molecular docking of estradiol to ER*α* shows the best binding affinity score of −9.63 kcal mol^−1^ (Table 1 and Appendix A), weaker than that of the ER*α*-4OHT complex. 

For the onoceranoid triterpenes, molecular docking results suggest that all display stronger binding energy scores than estradiol, except **1** (Table 1 and Appendix A). Among them, only **3** surpasses the binding strength over 4OHT, where the major cluster contains seventy docking poses (Appendix A), with the best binding energy of −11.60 kcal mol^−1^ (Table 1). Its strong binding is mainly owing to its intermolecular energy (−12.49 kcal mol^−1^, Table 1), which comes from one conventional H-bond and two hydrophobic alkyl–alkyl interactions (Table 2 and Appendix A). The best binding energy scores of **2** and **4** are −11.43 and −11.05 kcal mol^−1^, respectively, which are close to that of 4OHT (Table 1). Both onoceranoid triterpenes have an unfavorable interaction with ER*α*, lowering their intermolecular energy values (Table 2 and Appendix A). Meanwhile, the binding energy score of **5** is slightly stronger than that of estradiol in binding to ER*α* (Table 1). Its intermolecular energy originates (Table 1) from two hydrophobic alkyl–alkyl interactions (Table 2 and Appendix A).

Notably, 4OHT occupies the estradiol binding site with a hydrophobic characteristic (Figure 2). Additionally, the tertiary amine group of tamoxifen protrudes outside (Figure 2), causing H12 to be in an antagonist conformation [40]. As portrayed in Figure 2, all onoceranoid triterpenes adopt binding modes like 4OHT, suggesting that they may act as antagonist ligands for ERα. However, since **2**, **3**, and **4** have binding energy scores close to 4OHT, we only considered these onoceranoid triterpenes potential ERα inhibitors.

### 2.2. Prediction of Drug-Likeness

In the next stage, we employed Lipinski’s rule of five (Ro5) to predict the drug-likeness properties of the onoceranoid triterpenes. This approach foresees whether the compounds are orally active drugs in humans but does not indicate their pharmacological properties [41]. The rule covers chemical properties, including molecular weight (MW), hydrogen bond acceptor (HBA) and donor (HBD), and logP [42].

The prediction results (Table 3) show that **2**, **3**, and **4** satisfy MW, HBA, and HBD criteria but not logP. Their molecular weight values are below 500 g mol^−1^. Therefore, they may have a good permeability across the intestine and membrane lipid bilayer. According to Ro5, compounds should possess hydrogen bond donors of less than five and hydrogen bond acceptors of less than ten to be permeable to the lipid bilayer membrane, which is fulfilled by the three onoceranoid triterpenes (Table 3). For the last criterion, **2**, **3**, and **4** show logP values above 5 (Table 3), suggesting their poor absorption into the bloodstream. Since the number of violations is only one, all onoceranoid triterpenes (**2**, **3**, and **4**) fulfill drug-likeness criteria [41].

### 2.3. In Silico Absorption, Distribution, Metabolism, Excretion, and Toxicity (ADMET)

Subsequently, we subjected the three onoceranoid triterpenes to pharmacokinetic prediction, including absorption, distribution, metabolism, and excretion properties. These pharmacokinetic criteria are essential in drug development, as they provide insights into the behavior of compounds in the body and their potential efficacy [43]. Additionally, the toxicity properties of the onoceranoid triterpenes were assessed, which is crucial for evaluating their safety and possible adverse effects [44].

The absorption criterion is essential in drug development since it determines the extent and rate at which a drug enters systemic circulation, directly impacting its bioavailability and therapeutic efficacy [45]. The absorption criterion is predicted based on water solubility, Caco2 permeability, human intestinal absorption, P-glycoprotein substrate, and inhibitor [46]. The three onoceranoid triterpenes are predicted to have poor water solubility since their logS values are less than −4 log mol L^−1^ (Table 4). These results are anticipated due to their high portions of hydrophobic moieties (Figure 1) and consistent with their high logP values (Table 3). Nevertheless, the three onoceranoid triterpenes may readily cross the human intestinal mucosa as their predicted values of Caco2 permeability and human intestinal absorption are above 0.90 and 30%, respectively (Table 4). For the absorption criterion related to P-glycoprotein, **4** is predicted as a substrate, but **2** and **3** are not, suggesting that **4** is potentially pumped back to the lumen of the small intestine. However, the three onoceranoid triterpenes are predicted as P-glycoprotein I and II inhibitors, which prevent these proteins from forcing xenobiotic compounds back to the lumen [44].

After being absorbed through the intestinal wall, the three onoceranoid triterpenes will be distributed via the bloodstream from target organs or tissues. During this process, they may interact with proteins in the blood and membranes of various tissues. Key parameters used to predict drug distribution include the steady-state volume of distribution (VDss), fraction unbound, blood–brain barrier (BBB) permeability, and central nervous system (CNS) permeability [44]. VDss predictions (Table 4) suggest that **2** is more distributed in tissue than the blood plasma, log VDss > 0.45 [46], while **3** and **4** are moderately more distributed in tissue than the blood plasma, log VDss between −0.15 and 0.45 [46]. Thus, the administration of **2** requires more doses than the administration of **3** and **4**. Moreover, the three onoceranoid triterpenes are predicted to bind serum proteins, indicated by the fact that their fraction unbound values are 0 (Table 4), lowering their efficiency in diffusing cellular membranes. However, they could also moderately penetrate the blood–brain barrier since their log BB values are between −1.0 and 0.3, and **4** may more easily cross the barrier due to its log PS, which is greater than −2.00 (Table 4). 

In terms of metabolism, the three onoceranoid triterpenes are not predicted as substrates or inhibitors for the cytochrome P450 (CYP450) family, except as CYP3A4 substrates (Table 4). These xenobiotic compounds could be metabolized by CYP3A4. However, further prediction, using http://biotransformer.ca (accessed on 23 July 2023), suggests that they may not metabolized by CYP450, but **4** may undergo glucurodination. 

For the excretion criterion, the three onoceranoid triterpenes are predicted to have low total clearance values (Table 4), reflecting their slow clearance from the human body via kidney or hepatic excretion. The subsequent prediction suggests that they are not renal organic cation transporter 2 (OCT2) substrates. Therefore, they may be cleared from the human body through hepatic excretion. 

The assessment of toxicity is crucial in drug development to mitigate the risk of candidate failures. Thus, toxicity assessment using in silico approaches is beneficial for new drug candidates. In the case of the three onoceranoid triterpenes, they are predicted to be free from AMES toxicity, indicating a potential absence of carcinogenicity. Additionally, they may not cause hepatoxicity. However, it is worth noting that **2** and **3** are predicted to be hERG II inhibitors according to the pkCSM web server [46], raising concerns regarding their potential impact on cardiac safety. Nevertheless, according to the toxCSM [47], these compounds are classified at a medium safety level for cardiac effects. Therefore, the in silico ADMET evaluations conducted suggest that the three onoceranoid triterpenes hold promise as ER*α* inhibitor candidates.

### 2.4. Molecular Dynamics Simulations

Molecular docking is extensively employed in structural-based virtual screening because of its rapid evaluation [44]. Our study implemented molecular docking to screen onoceranoid triterpenes from *L. domesticum* Corr. cv. *kokosan*. However, molecular docking uses a rigid receptor, which does not account for the flexibility of the protein–ligand complex [36,44]. This rigidity can restrict the sampling of both ligand and receptor conformations, potentially affecting pose prediction accuracy. Additionally, molecular docking often employs simplified scoring functions, which may not correlate well with experimental binding affinities. These approximated scoring functions can produce results that do not accurately reflect the true binding affinities of compounds [48]. Hence, we subjected **2**, **3**, and **4** to molecular dynamics (MD) simulations, which accommodate receptor and ligand flexibility [49]. 

#### 2.4.1. The Effect of Ligand Binding to ERα Conformational Stability

The binding of any ligand to a protein, like ER*α*, may affect the conformational changes in both ligand and protein [49,50]. Conformational changes in the whole protein structure can be identified through analyses of RMSD and hierarchical clustering on principal components (HCPC) from molecular dynamics (MD) simulations [51]. As shown in Figure 3, ER*α* in its apo form is relatively stable after 10 ns. Its RMSD values remain between 3.00 and 4.00 Å, with median and interquartile range (IQR) values of 3.28 and 0.32 Å, respectively (Appendix A). According to HCPC, the apo form of ER*α* has five conformation clusters over the 500 ns trajectory (Appendix A). The two first conformations transiently appear in the beginning 10 ns, where RMSD values climb their hill (Figure 3: apo in red and yellow colors). Next, the third conformation (green) lasts until 100 ns. For the remaining 400 ns, the fourth (blue) and fifth (magenta) conformations are alternately present, with the fifth one as dominant. 

The binding of some ligands studied here can relatively stabilize the conformation of ER*α*. The most striking effect is exhibited by **2**. Its binding reduces protein RMSD values to a median of 2.46 Å and an IQR of 0.23 Å (Appendix A). The binding of **2** stabilizes ER*α* to maintain a relatively consistent conformation for the last 400 ns, even though in the first 100 ns, the protein exhibits intensive conformational changes (Figure 3 and Appendix A). The binding of 4OHT and estradiol also stabilizes ER*α* in certain conformations for around 300 ns (Figure 3 and Appendix A). Nevertheless, tamoxifen stabilizes ER*α* conformation more than estradiol. Based on our simulations, tamoxifen-bound ER*α* shows RMSD values with median and IQR of 2.94 and 0.26 Å, respectively, whereas estradiol-bound ER*α* has higher median (3.02 Å) and IQR (0.54 Å) values of protein RMSD. Moreover, estradiol-bound ER*α* shows some conformational changes in different clusters at the beginning and the end of the simulation (Figure 3 and Appendix A). 

The binding of **3** alters the conformation of ERα, but protein stability is comparable to the apo form. Its binding reduces RMSD values of ERα with a median of 2.66 Å, despite having a higher IQR (0.36 Å) than that of the apo form (0.32 Å) (Appendix A). HCPC suggests six conformation clusters of ERα throughout the 500 ns MD simulation (Figure 3 and Appendix A). Four different clusters alternately appear in the first 125 ns. For the next 115 ns, **3**-bound ERα is in the fifth cluster conformation (Figure 3 and Appendix A: row 3 in turquoise color). Subsequently, the sixth cluster conformation (Figure 3 and Appendix A: row 3 in magenta color) alternately emerges with the fifth one and consistently presents in the last 140 ns of MD simulation. Only the binding of **4** reduces the conformational stability of ERα (Figure 3A and Appendix A). During the 500 ns MD simulation, **4**-bound ERα alternately alters its conformation in sixteen different clusters. Most clusters transiently appear throughout the simulation. The RMSD values of each ligand were also analyzed. The results show that all of the ligands have low RMSD profiles, below 2.5 Å, indicating the stability of binding poses (Figure 3B). 

While RMSD describes conformational changes in the entire ERα structure over time, RMSF indicates the flexibility of specific residue during MD simulations. As depicted in Figure 4A,B, the apo and ligand-bound forms ERα fluctuate, particularly at H1, coil2-3, and the tail of H12. H1 is important for the stability of the receptor–DNA complex [37], while coil2-3 is important for the coactivators and corepressors [38]. Estradiol-bound ERα exhibits the highest flexibility at H1, whereas the one with tamoxifen is the lowest at this part. At the beginning of coil2-3, **3**- and **4**-bound ERα proteins have higher flexibilities than the others, but at the end of this coil, the highest flexibility is shown by the apo form of ERα. Interestingly, both tamoxifen- and estradiol-bound ERα proteins are the least fluctuated at the whole part of coil2-3. At the coil of H12, all ligand-bound and apo forms of ERα display comparable flexibility, but for the end of H12, **4**-bound ERα has the highest flexibility. H12 plays a role in regulating the activity of ERα by switching between active and inactive conformations [39]. This segment is the most flexible in all ligand-bound and apo forms of ERα (Figure 4A,B). 

We also performed HCPC on RMSF values of all ligand-bound and apo forms of ERα (Figure 5). The result suggests that tamoxifen- and **2**-bound ERα complexes are in the same cluster, suggesting their RMSF profile similarity. Interestingly, both complexes share the same parent branch with the estradiol-bound ER*α*, with a dissimilarity height of 36.5. They have strikingly different RMSF values at H1 and slightly different ones at H2, Coild2-3, H9, H10, and H12 (Appendix A). Meanwhile, the other ERα complexes are each in a separate cluster. Nevertheless, **3**-bound ER*α* and the apo form have the same parent branch, with a dissimilarity height of 36.6. 

#### 2.4.2. Molecular Mechanic Generalized Surface Area (MMGBSA) Binding Energy

From the obtained 500 ns MD trajectories, we calculated MMGBSA binding energy values every 10 ns window. As shown in Appendix A, MMGBSA binding energy values fluctuate for every ligand. Tamoxifen exhibits the highest fluctuated values (median absolute deviation (mad) = −3.96 kcal mol^−1^), while estradiol has the lowest fluctuated values (mad = −1.73 kcal mol^−1^). 

Nonetheless, MMGBSA binding energy values overlap between ligands, confusing further analysis. Thus, we performed a Kruskal–Wallis rank-sum test, followed by the Games–Howell test. The results suggest that **2** (median = −56.10 kcal mol^−1^), **3** (median = −52.27 kcal mol^−1^), and **4** (median = −52.17 kcal mol^−1^) are stronger than estradiol (−46.35 kcal mol^−1^) in binding to ER*α* (Figure 6 and Appendix A). Compound **2** binds stronger to ERα than **3,** which is in contrast to molecular docking results, according to which **3** binds stronger than **2** (Table 1). Additionally, compounds **2** and 4OHT (median = −57.37 kcal mol^−1^) have close median values of MMGBSA binding energy. These results are consistent with our in vitro experiment that **2** and 4OHT exhibited inhibition activities to MCF-7, with IC_50_ 29.73 µg mL^−1^ (unpublished result) and 20.5 µg mL^−1^, respectively. Therefore, **2** is the most potent ER*α* inhibitor.

#### 2.4.3. MMGBSA Binding Energy Decomposition

Subsequently, we performed MMGBSA binding energy decomposition to investigate the contributions of amino acid residues at the ERα binding site. In the MMGBSA binding energy decomposition analysis, Leu525, Leu346, and Ala350 are found to be highly contributed residues in binding all ligands to ER*α* (Figure 7), primarily through van der Waals and nonpolar solvation energy terms (Appendix A). Moreover, Glu353, Leu391, and His524 are identified as residues with stronger contributions in estradiol binding to ER*α*. However, certain residues, including Tyr526, Asp351, Ser395, Glu419, Asp426, Glu385, Gly344, and Glu330, exhibited unfavorable interactions with estradiol. These residues have median values of binding energy ranging from 0.03 to 0.15 kcal mol^−1^, which may explain the weak binding of estradiol compared with tamoxifen and **2**, **3**, and **4**.

In the case of 4OHT, Asp351, and Glu353, the additional residues provide binding energy values of −6.81 and −6.09 kcal mol^−1^, respectively. Such high binding energy contributions are from electrostatic interactions (Appendix A). Nonetheless, some residues cause unfavorable interactions with tamoxifen, including Ser395, Tyr526, Ala405, Lys416, Asp426, and Leu541. These residues exert repulsive energy with median values ranging from 0.02 to 0.08 kcal mol^−1^. 

While no residue other than Leu525, Leu346, and Ala350 provides binding energy stronger than −3.00 kcal mol^−1^ for the binding of **4** to ER*α*, Thr347 provides binding energy with median values of −3.69 kcal mol^−1^ for **3**. Like estradiol and tamoxifen, the binding of **3** to ER*α* involves repulsive binding caused by Glu419 and Gly344, with median values of 0.02 and 0.03 kcal mol^−1^, respectively. The binding of **4** to ER*α* also includes unfavorable interactions contributed by Asp426, Ser518, and Glu385, with median binding energy values ranging from 0.02 to 0.04 kcal mol^−1^. Meanwhile, the binding of **2** to ER*α* involves two additional residues, Thr347 and Met528, with strong binding energy, where the median values are −4.35 and −3.39 kcal mol^−1^, respectively. The binding is also not free from repulsive interactions, which are contributed by Asp426, Ser395, Glu330, and Glu419, with median binding energy values ranging from 0.01 to 0.08 kcal mol^−1^. 

#### 2.4.4. Nonbonded Interactions 

Although **2**, **3**, and **4** show stronger binding energy values than estradiol in binding ER*α*, they only form transient conventional H-bond with the receptor (Appendix A). Furthermore, the highest H-bond occurrence for **2**, which more tightly binds ER*α* than **3** and **4**, is only 0.05% throughout the 500 ns trajectory. In contrast, estradiol establishes H-bond interactions with His524 and Glu353, with conservation of 40.31% and 39.11%. These results indicate the binding of **2**, **3**, and **4** to ER*α* is owing to nonpolar interactions, particularly van der Waals forces (Appendix A), which are also shown by MMGBSA binding energy contribution (Appendix A). 

Van der Waals forces occur when atoms are very close, and their outer electron clouds barely touch. These interactions create fluctuations in charge that cause a weak attraction without any specific direction [52]. They also contribute to the binding energy calculation, which depends on the contact surface area (CSA) between the residues in the binding site and the respective ligand. To further explain the different affinities of each ligand toward ER*α*, we analyzed the probability of each ligand interacting with the binding site’s residues by mapping their CSA. As shown in Figure 8 and Appendix A, estradiol, the agonist ligand, has the lowest CSA values since its interaction involves fewer residues than other ligands. The antagonist ligand, 4OHT, has higher CSA values, followed by compounds **4**, **3**, and **2** (Appendix A). Moreover, **3** and **2** have a rather similar CSA map, which is understandable considering the fact that their structure is only different by the double-bond position. Our present work involves an enhanced method, implemented in a dr_sasa program, that calculates an atom’s contact area with neighboring atoms by directly assessing overlapping surface sections and considering potential shielding from nearby atoms [53].

The CSA map can also confirm that the residues Thr347 and Leu525 have important roles in the binding of onoceranoid triterpenes, which is shown by the high CSA values around 30 to 50 Å^2^ (Appendix A). In the compound **2** system, residue Thr347 has higher CSA values than Leu525 since the widest site of the residue Thr347 surface complements the ligand surface properly and facilitates the van der Waals interaction, rather than the edge part of residue Leu525 only, as shown in Figure 9. These results are in agreement with a higher MMGBSA binding energy contribution of Thr347 than Leu525. Furthermore, these results are consistent with a previous study on the boiling points of alkanes and cycloalkanes indicating that Van der Waals forces are influenced by the size and shape of molecules. When the surface area is smaller, there is less interaction between neighboring molecules, leading to weaker van der Waals forces [54]. 

The same high CSA values pattern was also found in residues Leu346 and Leu387, although their contribution to the binding energy is not as notably high as the key residues. In addition, the pattern in the CSA map of the three onoceranoid triterpenes (**2**, **3**, and **4**) is more similar to that of 4OHT, compared with estradiol, supporting our notion that these compounds will likely be the antagonist ligands similar to 4OHT. Studies showed that the majority (over 60%) of effective anti-cancer medications currently utilized in clinical practice are sourced from natural products found in plants, marine life, and microorganisms [55]. Notably, compounds like alkaloids, flavonoids, terpenoids, polysaccharides, saponins, and various others are recognized as naturally occurring bioactive substances with strong anti-cancer capabilities [17,56,57,58]. Thus, exploring the characteristics of bioactive compounds such as the one from *L. domesticum* Corr. cv. *kokosan* would provide new insight into the discovery and development of cancer drugs.

## 3. Materials and Methods

### 3.1. Preparation of Onoceranoid Triterpenes Structures

The BIOVIA Draw 2017 software version 17.1 (Dassault Systèmes, San Diego, CA, USA) was utilized to draw the 2D structures of onoceranoid triterpenes manually. MarvinView 18.21.0 (ChemAxon, Sydney, NSW, Australia) was employed to predict the protonation states of the onoceranoid triterpenes under physiological pH conditions (pH 7.4). BIOVIA Discovery Studio 2021 Visualizer v21.1.0.20298 (Dassault Systèmes, San Diego, CA, USA) was utilized to transform 2D into 3D structures in a pdb format.

### 3.2. Molecular Docking of Onoceranoid Triterpenes against ERα

Molecular docking was validated by redocking the 4OHT structure into the ER*α* binding site. The procedure was deemed successful if the root-mean-square deviation (RMSD) value for the docked 4OHT was less than 2.00 Å [44]. Briefly, the 3D structure of human ER*α* binding 4OHT (3ERT) [40] was obtained from the Protein Data Bank (PDB). ER*α* and 4OHT structures were separately saved as individual pdb files, with water and ion molecules excluded. Next, each file was subjected to molecular docking preparation in AutoDockTools 1.5.6 [59] (The Scripps Research Institute, La Jolla, CA, USA). The preparation included hydrogen atom addition; Kollman and Gasteiger atomic charge assignment to ER*α* and 4OHT, respectively; nonpolar hydrogen removal; and conversion to pdbqt format. The active torsions in the 4OHT structure were configured based on recommendations from AutoDockTools 1.5.6. Grid parameters were initiated from the center of 4OHT, located at the ER*α* binding site. The grid box size was 40 × 40 × 40 with center coordinates of 30.010, −1.913, 24.207 (x, y, z), and 0.375 Å spacing. Grid maps for the ER*α* within the grid box were computed using AutoGrid4.2 [40] (The Scripps Research Institute, La Jolla, CA, USA). A Lamarckian genetic algorithm (LGA) was applied with 100 genetic algorithm (GA) runs, 300 population sizes, and 2,500,000 maximum evaluations. The remaining search and docking parameters were left at their default settings. The redocking procedure was performed using AutoDock4.2 [40] (The Scripps Research Institute, La Jolla, CA, USA). 

The onoceranoid triterpene structures were prepared in AutoDockTools 1.5.6 using the procedures outlined in the molecular docking validation. Docking results were sorted based on binding free energy and histogram groups. The pose with the lowest energy value was selected for further interaction analysis with ER*α*, which was visualized using BIOVIA Discovery Studio 2019 Visualizer.

### 3.3. Lipinski’s Rule of Five and ADMET Analysis 

The SwissADME web server [60] was used to calculate Lipinski’s rule of five (Ro5). Ro5 was determined to assess the potency of onoceranoid triterpenes as orally active drug candidates. Its assessment is based on molecular weight (MW), *n*-octanol–water partition coefficient (logP), and number of hydrogen bond acceptors (HBA) and donors (HBD).

In silico absorption, distribution, metabolism, excretion, and toxicity assessments were carried out using the pkCSM web server [46]. The criteria for absorption included factors such as water solubility, permeability in Caco2, and human intestines, and whether they were substrates or inhibitors for P-glycoprotein I and II. For distribution predictions, we took into account the steady-state volume of distribution (VDss), the unbound fraction, and the permeabilities of the blood–brain barrier (BBB) and central nervous system (CNS). Metabolism assessments focused on the potential inhibition of CYP1A2, CYP2C19, CYP2C9, and CYP2D6 while predicting substrates for CYP2D6 and CYP3A4. Total clearance and the role of renal organic cation transporter 2 (OCT2) were considered to gauge excretion. For safety assessment, predictions were made regarding AMES toxicity and hepatoxicity using pkCSM. The web server was also utilized to estimate the efficacy of hits as inhibitors for hERG I and II. Additionally, for toxicity predictions, we also employed the toxCSM web server [47].

### 3.4. Molecular Dynamics Simulations

Initially, we prepared partial charges of onoceranoid triterpenes and 4OHT structures using Austin Model 1—Bond Charge Corrections (AM1-BCC), as implemented in the antechamber program, AmberTools21 [61] (AMBER, San Francisco, CA, USA). The other parameters for onoceranoid triterpenes and 4OHT structures were obtained from Generalized Amber Force Fields 2 (GAFF2). ER*α* structure was prepared using the pdb4amber program, AmberTools21, to adjust histidine residues according to their local chemical environments. We assigned ff19SB [62] for the ER*α* structure. The tleap program, AmberTools21, was used to prepare each ER*α* and ligand complex. The TIP3P explicit water model was employed to solvate every ER*α*–ligand complex, with a minimal boundary box of 10 Å. Moreover, we introduced a few Na^+^ and Cl^−^ ions to achieve a physiological salt concentration of 0.15 M.

Molecular dynamics simulation of every ER*α* system was performed using Particle-Mesh Ewald Molecular Dynamics (PMEMD) on AMBER20, employing GPU acceleration [61]. Initial minimization encompassed 1000 steps of steepest descent and 2000 steps of conjugate gradient, applying 5 kcal mol^−1^ Å^−2^ harmonic force. Subsequently, 5000 steps of unimpeded conjugate gradient minimization rectified spatial overlaps. The system temperature ramped to 300 K in 20 ps increments (0–100 K; 100–200 K; and 200–300 K) over 60 ps. Equilibration ensured density, pressure, and gradual force release during 1000 ps. The production runs for 500 ns with a 2 fs time step. 

### 3.5. Trajectory Analysis

The *ccptraj* program in AmberTools21 was employed for MD trajectory analyses. The analyses include root-mean-square deviation (RMSD), root-mean-square fluctuation (RMSF), and conventional H-bond conservation. Molecular dynamics simulation trajectories were assessed using VMD. The ante-MMPBSA.py in AmberTools21 was used to calculate MMGBSA binding energy and decomposition analysis. To compute contact surface analysis (CSA), we used the dr_sasa program [53]. Jupyter Notebook 6.4.7 (Project Jupyter, Berkeley, CA, USA) [63] was used to conduct analyses under an R programming language environment version 3.6.1 (R Foundation for Statistical Computing, Vienna, Austria) [64]. Statistical analyses were performed using an R package, rstatix [65]. Hierarchical clustering on principal analysis (HCPC) for RMSD was carried out using an R package, bio3d [66], while for RMSF, FactoMineR [67] and factoExtra [68] were used. Graphs were generated using R packages tidyr [69], ggplot2 [70], and ggpubr [71]. Inkscape 1.3 (The Inkscape Project, Boston, MA, USA) [72] was used to create the visual illustrations. 

## 4. Conclusions

In the present study, we evaluated the anti-breast cancer potential of five onoceranoid triterpenes found in *Lansium domesticum* Corr. cv. *kokosan* against estrogen receptor alpha (ER*α*) using a range of in silico techniques, including molecular docking, Lipinski’s rule of five, in silico ADMET, and molecular dynamics simulations. The potential onoceranoid triterpenes based on molecular docking—namely 8,14-secogammacera-7,14-dien-3,21-dione (**2**); 8,14-secogammacera-7,14(27)-dien-3,21-dione (**3**); and kokosanolide B (**4**)—fulfill drug-likeness criteria and potential ADMET profiles. Molecular dynamics simulations revealed **2** as a highly promising contender, exhibiting the most favorable MMGBSA binding energy of −56.10 kcal mol^−1^, closely followed by **3** and **4,** with MMGBSA binding energy values of −52.27 and −52.17 kcal mol^−1^, respectively. These computational findings align well with our experimental data, where **2** demonstrated notable inhibition activity against MCF-7 cells, with an IC_50_ of 29.73 µg mL^−1^, while tamoxifen exhibited an IC_50_ of 20.5 µg mL^−1^. Consequently, **2** emerges as the most potential inhibitor of ERα. HCPC analysis on RMSF suggests that **2** and 4OHT are in the same cluster. Moreover, the spatial distribution of CSA among the three onoceranoid triterpenes (**2**, **3**, and **4**) bears greater similarities to that of 4OHT than estradiol. This observation suggests a hypothesis that these compounds may potentially function as antagonist ligands akin to 4OHT. These findings collectively provide valuable insights into the potential therapeutic significance of these triterpenes in targeting ERα-associated breast cancer. 

## Figures and Tables

**Figure 1 ijms-24-15033-f001:**
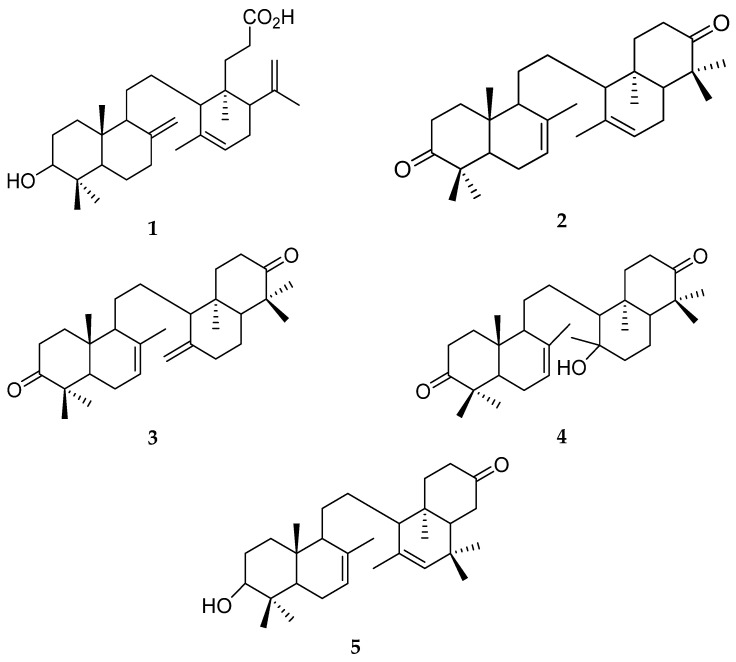
Five onoceranoid triterpenes from *Lansium domesticum* Corr. cv. *kokosan*: lansiolic acid (**1**) [31], 8,14-secogammacera-7,14-dien-3,21-dione (**2**) [32], 8,14-secogammacera-7,14(27)-dien-3,21-dione (**3**) [32], kokosanolide B (**4**) [27], and 3-hydroxy-8,14-secogamasera-7,14-dien-21-one (**5**) [33].

**Figure 2 ijms-24-15033-f002:**
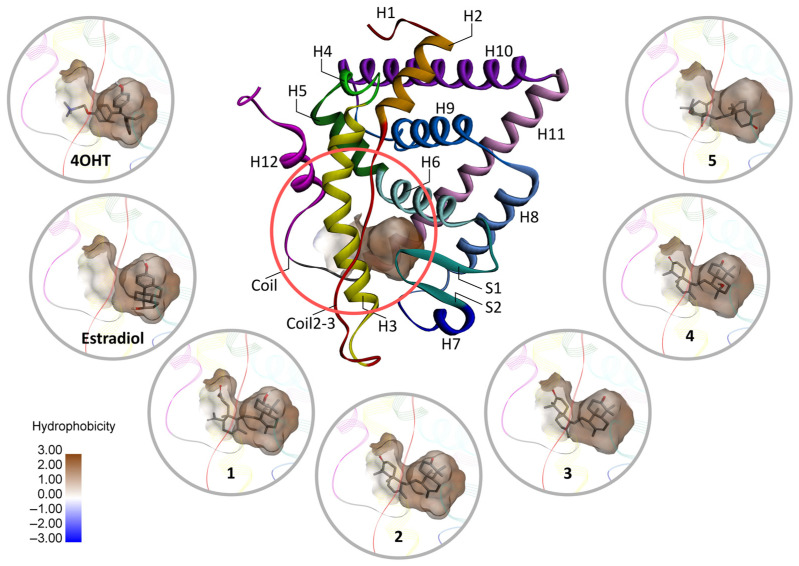
Docking poses of ligands in the hydrophobic binding site of ER*α*. The docking poses were generated through molecular docking. The ligand binding domain of ER*α* consists of twelve helices (H1–H12), beta sheets **1** and **2** (S1, S2), coil separating H11 and H12, and coil2-3, which are in different colors. The hydrophobicity of the molecular surface is represented by brown to white and blue color coding. The brown and color scales denote hydrophobicity and hydrophilicity indices, respectively, whereas the white color shows the balance between both properties.

**Figure 3 ijms-24-15033-f003:**
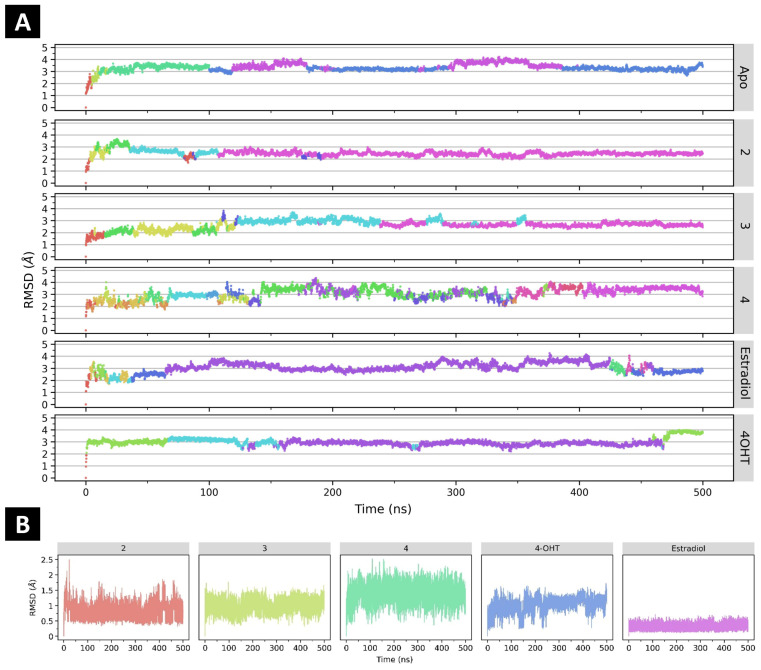
(**A**) The time evolutions of root-mean-square deviations (RMSDs) of ERα in apo and ligand-binding forms. The different colors represent the different clusters from the principal component analysis clustering (Appendix A). The color gradient is from red (cluster 1) to yellow (cluster 2), green (cluster 3), cyan (cluster 4), blue (cluster 5), and magenta (cluster 6). (**B**) RMSD analysis of each ligand during 500 ns MD simulations, calculated with the cpptraj program in Amber20. Data visualization was carried out using an R package ggplot2 on Jupyter Notebook 6.4.7. The different colors represent the different ligand systems.

**Figure 4 ijms-24-15033-f004:**
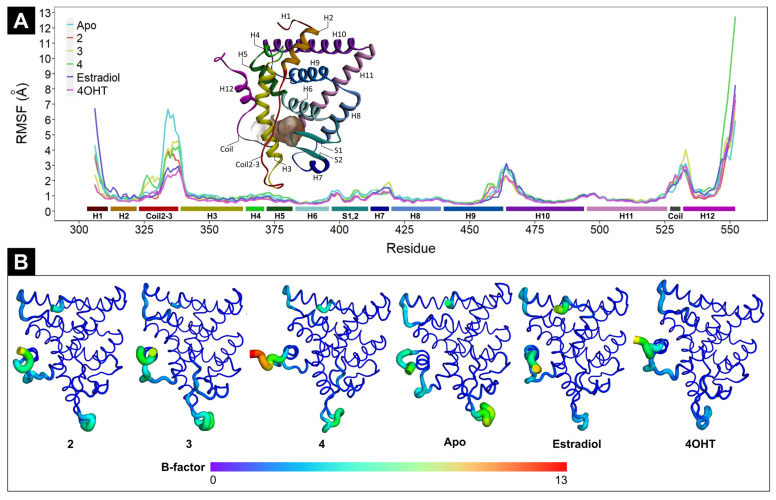
(**A**) RMSF plots of ERα in apo and ligand-bound forms. The turquoise line represents the RMSF profile of ERα in the apo form, the red line denotes **2**-bound ER*α*, the yellow line is **3**-bound ERα, the green line is **4**-bound ERα, the purple line is estradiol-bound ERα, and the magenta line is 4OHT-bound ERα. Colored bars under the RMSF lines correspond to ERα segments. (**B**) The 3D structures of apo and ligand-bound ERα in b-factor putty representations. The difference in each color represents the range of RMSF values in Angstrom Å. The color gradient is from purple (0 Å) to red (13 Å); the closer the color to the red, the higher the RMSF value.

**Figure 5 ijms-24-15033-f005:**
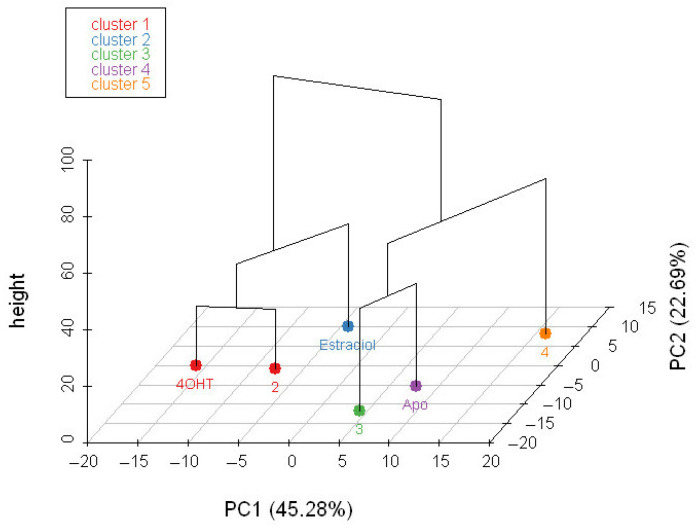
Hierarchical clustering on the factor map of apo and ligand-bound ER*α*. The clustering is based on the RMSF of apo and ligand-bound ER*α*. The different colors represent the different clusters from the principal component analysis clustering.

**Figure 6 ijms-24-15033-f006:**
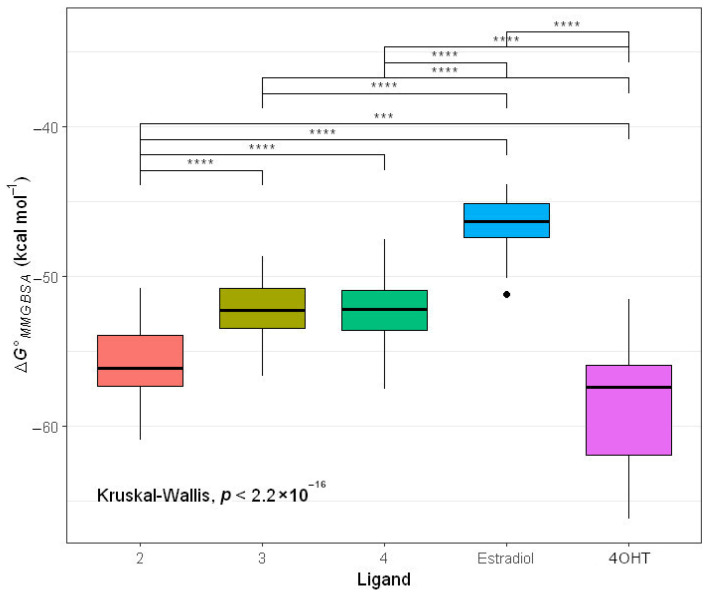
Box plots of MMGBSA binding energy for ligand binding to ER*α*. The Box plots are from data points in Appendix A. The Box plots are complemented with a *p*-value from the Kruskal–Wallis test and pairwise statistical significances from the Games-–Howell test, displayed by asterisk symbols. Asterisk symbols *** and **** represent *p*-values in magnitudes of 10^−4^ and less than 10^−4^, respectively. The box plots depict five summary statistics, including the minimum, first quartile, second quartile, third quartile, and maximum values, of MMGBSA binding energy values. A circled point indicates a potential outlier value. The various colors correspond to Box plots depicting MMGBSA binding energy values for the different ligands.

**Figure 7 ijms-24-15033-f007:**
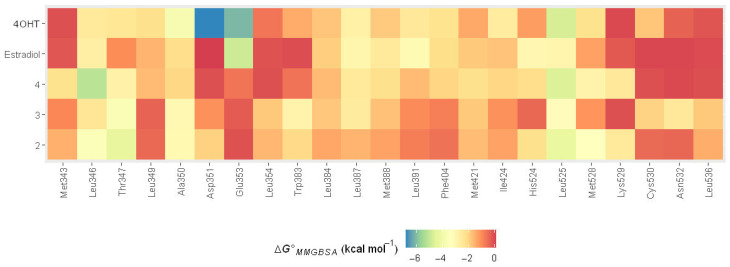
Heatmap of MMGBSA binding energy decomposition for ER*α* residues interacting with ligands. The difference in each color represents the range of ΔG^0^*_MMGBSA_* values in kcal.mol^−1^. The color gradient is from blue (around −7 kcal.mol^−1^) to red (around 0 kcal.mol^−1^); the closer the color to the blue, the better the binding affinity.

**Figure 8 ijms-24-15033-f008:**
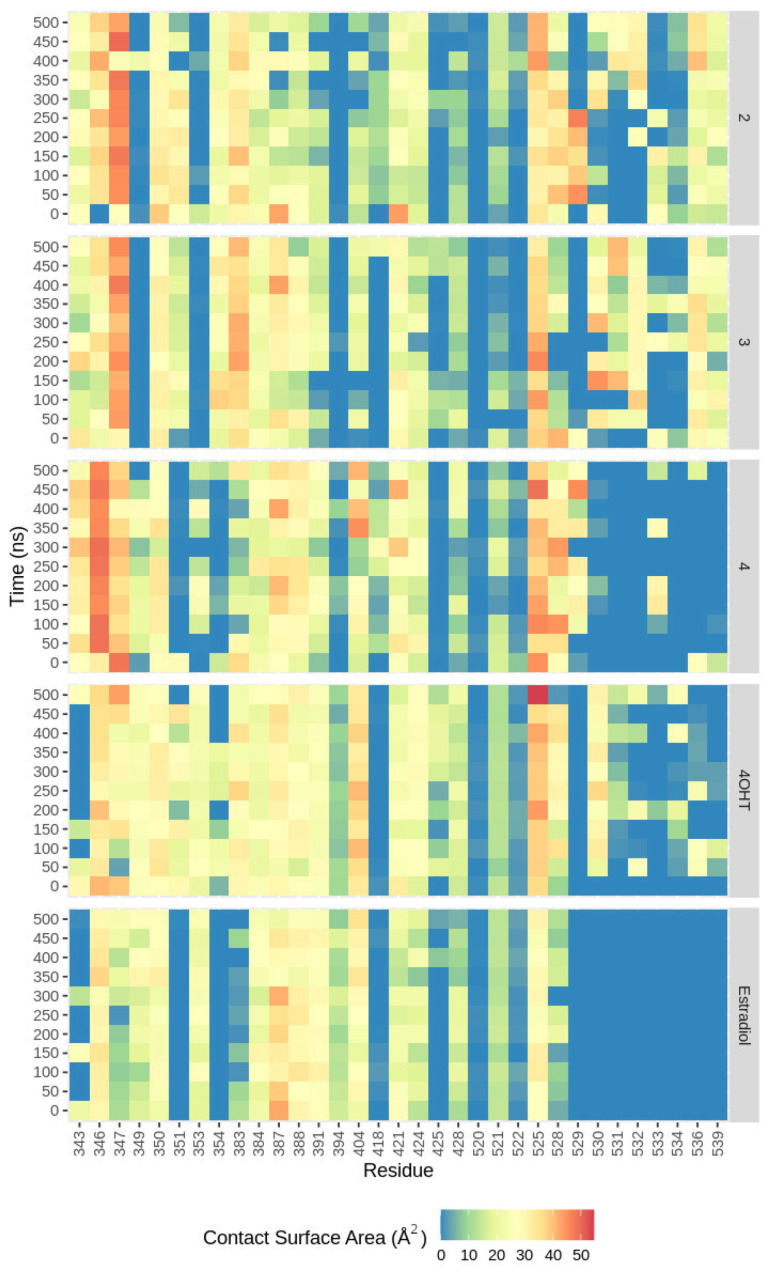
Contact surface area heatmap for each ligand interacting with ER*α* residues in the binding sites. The difference in each color represents the range of contact surface area in Å^2^. The color gradient is from blue (0 Å^2^) to red (55 Å^2^); the closer the color to red, the higher the area.

**Figure 9 ijms-24-15033-f009:**
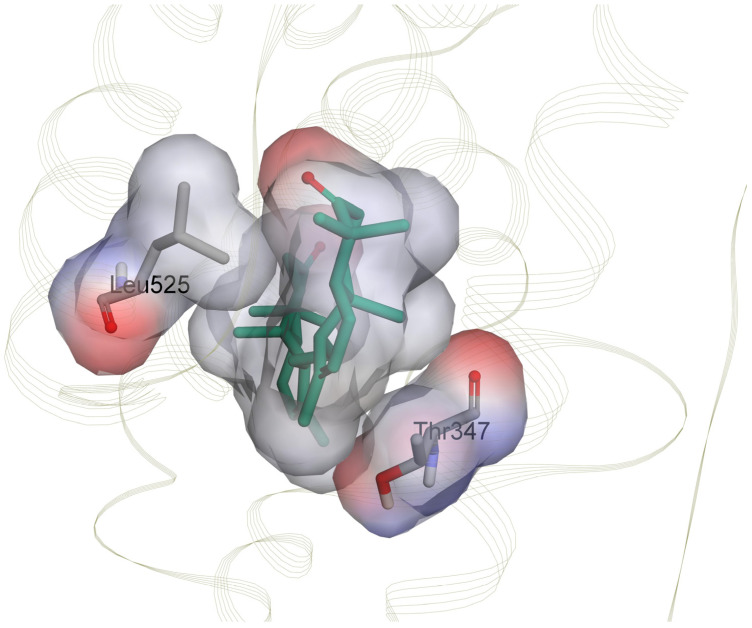
Van der Waals interaction of **2** with residues Thr347 and Leu525 of ER*α.* The surface contacts are visualized using an atomic charge surface. Compound **2** is displayed as a green licorice structure. The different colors represent the average atomic charge. The red color represents the negative charge, the blue indicates the positive charge, and the greyish white denotes the uncharged atom.

**Table 1 ijms-24-15033-t001:** Binding energy scores and their components of tamoxifen, estradiol, and five onoceranoid triterpenes to ER*α*. The scores were generated from molecular docking using AutoDock4.

Ligand	Binding Energy (kcal mol^−1^)	Intermolecular Energy (kcal mol^−1^)	Torsional Energy (kcal mol^−1^)
4OHT	−11.50	−14.18	2.68
Estradiol	−9.63	−9.63	0
**1**	−9.37	−11.45	2.09
**2**	−11.43	−12.33	0.89
**3**	−11.60	−12.49	0.89
**4**	−11.05	−12.24	1.19
**5**	−10.03	−10.92	0.89

**Table 2 ijms-24-15033-t002:** Interactions between ligands and ER*α.* The interactions are based on molecular docking results.

Ligand	Attractive Charge	Conventional H-Bond	Carbon H-Bond	Hydrophobic		Unfavorable Interaction
				Alkyl	π–Alkyl	
4OHT	1	2	2	2	7	0
Estradiol	0	3	0	3	5	0
**1**	0	0	0	16	1	1
**2**	0	1	0	0	0	1
**3**	0	1	0	2	0	0
**4**	0	1	0	0	0	1
**5**	0	0	0	2	0	0

**Table 3 ijms-24-15033-t003:** Ro5 assessment results for three onoceranoid triterpenes and the reference ligands. MW: molecular weight, logP: octanol–water partition coefficient, HBA: hydrogen-bond acceptor, HBD: hydrogen-bond donor.

Compound	MW (g mol^−1^)	logP	HBA	HBD	Ro5 Violations
Estradiol	272.38	3.60	2	2	0
4OHT	371.52	5.99	2	0	1
**2**	452.72	8.11	2	0	1
**3**	452.72	8.11	2	0	1
**4**	470.73	7.30	3	1	1

**Table 4 ijms-24-15033-t004:** In silico absorption, distribution, metabolism, excretion, and toxicity (ADMET) assessment on three onoceranoid triterpenes. The assessment was performed using the pkCSM web server [46].

Criterion	Onoceranoid Triterpene		
	2	3	4
Absorption			
Water solubility (log mol L^−1^)	−7.21	−7.18	−6.55
Caco2 permeability (log cm s^−1^)	1.22	1.24	1.21
Human intestinal absorption (%)	96.41	97.33	95.01
P-glycoprotein substrate	No	No	Yes
P-glycoprotein I inhibitor	Yes	Yes	Yes
P-glycoprotein II inhibitor	Yes	Yes	Yes
Distribution			
VDss in humans (log L kg^−1^)	0.5	0.33	0.3
Fraction unbound	0	0	0
BBB permeability (log blood–brain barrier permeability)	0.14	0.04	−0.20
CNS permeability (log blood–brain permeability surface) area	−2.53	−2.25	−1.23
Metabolism			
CYP2D6 substrate	No	No	No
CYP3A4 substrate	Yes	Yes	Yes
CYP1A2 inhibitor	No	No	No
CYP2C19 inhibitor	No	No	No
CYP2C9 inhibitor	No	No	No
CYP2D6 inhibitor	No	No	No
CYP3A4 inhibitor	No	No	No
Excretion			
Total clearance (log mL min^−1^ kg^−1^)	0.37	0.29	0.24
Renal OCT2 substrate	No	No	No
Toxicity			
AMES toxicity	No	No	No
hERG I inhibitor	No	No	No
hERG II inhibitor	Yes	Yes	No
Hepatotoxicity	No	No	No

## Data Availability

Data can be found in this article and the Appendix A.

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
