# Peer review of "Unveiling the Anti-Cancer Potential of Onoceranoid Triterpenes from Lansium domesticum Corr. cv. kokosan: An In Silico Study against Estrogen Receptor Alpha"

_ijms, 2023, doi:10.3390/ijms241915033_

Round 1

Reviewer 1 Report

September 10, 2023

Manuscript Number: ijms-2600030

Title: Unveiling the Anti-Cancer Potential of Onoceranoid Triterpenes from Lansium domesticum Corr cv Kokosan: In Silico Study Against Estrogen Receptor Alpha

Authors: Ari Hardianto, Sarah Syifa Mardetia, Wanda Destiarani, Yudha Prawira Budiman, Dikdik Kurnia, and Tri Mayanti

This manuscript reports the results of a computational evaluation of the anti-breast-cancer potential of five onoceranoid triterpenes found in Lansium domesticum (kokosan) against estrogen receptor alpha. Using various in silico techniques such as molecular docking, Lipinski's rule of five, in silico ADMET, and molecular dynamics simulations, the authors systematically assessed the potential from different aspects. Finally, it was proposed that one of onoceranoid triterpenes considered, 8,14-secogammacera-7,14-dien-3,21-dione (2), acts similarly to the ER native ligands, tamoxifen (4OHT) and estradiol. The computational methods used are not specific, but the analysis is very well done. Although this work is purely computational, but it appears that the authors also performed in vitro experimental analysis (unpublished). The manuscript is well written, the data are clearly presented to draw the conclusions. Therefore, this work is suitable for the publication in Int. J. Mol. Sci.. I have only minor comments to improve the manuscript.

1) “Our experimental data” on page 11 and 16 should be explained in more detail if possible. The comparison with experimental data is important not only to validate the results but also to distinguish this work from many other in silico papers without experimental validation. Additionally, it would be desired if compounds 3 and 4 were also compared to the experiments.

2) Ligands’ RMSDs can also be given to see the stability of binding poses.

3) In Figure 3, the caption should explain the colors used in the plot. These colors appear to correspond to the conformation of the cluster, but there is no explanation in the caption. The same is true for Figure 5, and furthermore, the author should also explain the mode of PC, perhaps using arrows on the 3-D structure. 

Reviewer 2 Report

In the present manuscript, titled “Unveiling the AntiCancer Potential of Onoceranoid Triterpenes from Lansium domesticum Corr cv Kokosan: In Silico Study Against Estrogen Receptor Alpha”, Hardianto and co-workers evaluated the potential of five onoceranoid triterpenes from in Lansium domesticum (kokosan) against estrogen receptor alpha (ERα) using in silico techniques, such as molecular docking, Lipinski's rule of five, in silico ADMET, and molecular dynamics simulations.

In reviewed manuscript selected paragraphs require some correction. I believe after some simple changes this paper is viable for publication and I will list my comment below. For this reason, I recommend publication after minor revision.

1.          The "Methods" section does not provide any info, or very little, about the definition of the docking grid, the optimization of the docking parameters, and the ligands/protein preparation. It is suggested to improve this section;

2.          SwissADME could also investigate a set of well consolidated parameters for searching bioactive compounds, such as PAINS filters, Lipinski’s rules, Veber, and Egan filters. If the authors did not do so, it is strongly suggested to perform them;

3.          I suggest investigating the RMSD of the protein;

4.          I suggest evaluating the Binding Energy across the simulation time for the examined complexes and compare them with the reference ligand;

5.          Evaluation of the protein/ligands interactions can be monitored throughout the molecular dynamic simulations. These interactions can be categorized type (such as Hydrogen Bonds, Hydrophobic, Ionic and Water Bridges) by and summarized in a plot above.  

Minor editing of English language required

Reviewer 3 Report

This is an example of well designed and performed in silico studies and overall a fine read.

My very minor suggestion is to add calculated binding poses visualisations for each of the tested ligands in the SI, along with their LID's.
